

# Simultaneous detection of atmospheric HONO and NO₂ utilizing an IBBCEAS system based on an iterative algorithm

Ke Tang[1,2], Min Qin[1], Wu Fang[1], Jun Duan[1], Fanhao Meng[1,2], Kaidi Ye[1,2], Helu Zhang[1,2], Pinhua Xie[1,2,3], Yabai He[1], Wenbin Xu[4], Jianguo Liu[1], and Wenqing Liu[1]

[1]Key Laboratory of Environment Optics and Technology, Anhui Institute of Optics and Fine Mechanics, Chinese Academy of Sciences, Hefei, 230031, China
[2]University of Science and Technology of China, Hefei, 230027, China
[3]CAS Center for Excellence in Urban Atmospheric Environment, Institute of Urban Environment, Chinese Academy of Sciences, Xiamen, 361021, China
[4]Science and Technology on Optical Radiation Laboratory, Beijing 100854, China

*Correspondence to*: Min Qin (mqin@aiofm.ac.cn)

**Abstract.** We present an improved incoherent broadband cavity-enhanced absorption spectroscopy (IBBCEAS) system based on iterative retrieval algorithm for simultaneous detection of atmospheric nitrous acid (HONO) and nitrogen dioxide (NO₂). The conventional IBBCEAS retrieval algorithm depends on the absolute change of light intensity, which requires high stability of light source and stable transmission of light intensity of all optical components. The new algorithm has an iterative module to obtain the effective absorption optical path length, and then the concentrations of HONO and NO₂ are determined by differential optical absorption spectroscopy (DOAS) retrieval, so it is insensitive to the fluctuation of absolute light intensity. The robustness of the system is verified by simulating the influence of the relative change of light intensity on the spectral retrieval results. The effect of nitrogen purging in front of the cavity mirrors on shortening the actual cavity length was measured and corrected by using NO₂ gas samples. Allan deviation analysis was conducted to determine the system stability, indicating the detection limits (2δ) of HONO and NO₂ are 0.08 ppbv and 0.14 ppbv at an integration time of 60 s, respectively. Furthermore, Kalman filtering was used to improve the detection precision of the system. The detection sensitivity of the system at an integration time of 3 s can be improved by 4.5-fold with applying Kalman filtering, which is equivalent to the detection sensitivity at an integration time of 60 s without applying Kalman filtering. The atmospheric HONO and NO₂ were observed by the IBBCEAS system based on iterative algorithm, and compared with those measured by conventional IBBCEAS.

## 1 Introduction

As HONO can absorb 300–400 nm solar radiation to form hydroxyl radical (OH) and nitric oxide (NO), it has been demonstrated that HONO contributes significantly to the OH budget during the daytime (Harrison et al., 1996). Recent studies have shown that the contribution of HONO to OH production plays an important role not only in the morning but also throughout the day (Spataro et al., 2013;Alicke, 2002). It is reported



that the contribution of HONO photolysis to OH production can reach 60% during the day (Michoud et al., 2012;Lu et al., 2013). However, the exact mechanisms leading to HONO formation are still under discussion. Existing gas-phase sources cannot explain the high concentration of HONO observed during the daytime (Zhou et al., 2002), indicating unknown sources of strong HONO during the day (Acker et al., 2006;Kleffmann, 2005). Therefore, fast and accurate measurement of HONO concentration is the premise of

studying the atmospheric chemical behavior of HONO and its contribution to regional oxidation. However, the lifetime of HONO may be only a few minutes with a low concentration during the day, even a few hundred parts-per-trillion-by-volume ($10^{-12}$, pptv) (Laufs et al., 2017;Hou et al., 2016). Therefore, a rapid and accurate detection of HONO has become a challenge.

HONO detection methods are mainly classified into two categories: One is based on wet chemical techniques,

and the other is based on spectroscopic methods. The wet chemical method mainly includes Denuder absorption coupling Ion Chromatography (Denuder-IC) (Neftel et al., 1996), Gas and Aerosol Collector (GAC) (Dong et al., 2012), Stripping Coil-Ion Chromatography (SC-IC) (Xue et al., 2019;Cheng et al., 2013), and Long Path Absorption Photometry (LOPAP) (Chen et al., 2014;Heland et al., 2001;Kleffmann et al., 2006). These kinds of methods have a lower detection limit and can reach several pptv. However, these

methods need to be calibrated to obtain accurate HONO concentration. Furthermore, the maintenance is cumbersome and requires frequent replacement of the chemical solution. Spectroscopic methods are based on the Lambert-Beer law and quantify the concentration of HONO by measuring the its absorption spectrum in a specific wavelength region, which is not easily affected by chemical interferences. Spectroscopic methods can be divided into conventional absorption methods and cavity-enhanced methods (Fiedler et al.,

2003). Conventional absorption methods mainly include Differential Optical Absorption Spectroscopy (DOAS) (Tsai et al., 2018;Qin et al., 2009), Fourier Transform Infrared Spectroscopy (FTIR) (Stockwell et al., 2014), and Infrared Quantum Cascade Laser (QCL) absorption spectroscopy (Cui et al., 2019). In these methods, the absorption spectrum of gas is obtained by the beam through multi-pass cells or long open path, the optical path length is a key factor that affects the sensitivity of the system. The cavity-enhanced methods

are based on high-finesse optical cavity-enhanced absorption spectroscopy, mainly including Cavity Ring Down Spectroscopy (CRDS) (Wang and Zhang, 2000) and Incoherent Broadband Cavity-Enhanced Absorption Spectroscopy (IBBCEAS) (Jordan et al., 2019;Duan et al., 2018;Gherman et al., 2008;Nakashima and Sadanaga, 2017;Donaldson et al., 2013;Scharko et al., 2014;Wu et al., 2014). IBBCEAS methods have higher spatial resolution and are easier to deploy on different platforms. Compared to CRDS techniques based

on a single wavelength, IBBCEAS can achieve simultaneous measurements of multi-gases by using a broadband light source. The IBBCEAS technology is such that the light beam is reflected back and forth in a high finesse optical cavity formed by two high-reflectivity mirrors. An optical cavity of several tens of centimeters can make the effective absorption path length reaching several kilometers, thereby improving the detection limit of the system. In recent years, IBBCEAS technology has been demonstrated to applied HONO

field measurements in remote (Duan et al., 2018;Tang et al., 2019) and urban regions (Crilley et al., 2019;Wu et al., 2014;Min et al., 2016;Nakashima and Sadanaga, 2017). Although a large number of HONO inter-





comparisons between different instruments have been carried out in previous studies (Crilley et al., 2019;Duan et al., 2018;Xue et al., 2019;Kleffmann et al., 2006;Stutz et al., 2010), deviations of HONO measurements between different methods still exist. In a recent field observation, the correlation between

different instruments is high ($r^2 > 0.97$), unfortunately, the absolute concentration difference even reaches 39% (Crilley et al., 2019). The factors that affect the accuracy of measurement need to be discussed further. Conventional IBBCEAS technology retrieves HONO concentration by measuring the absolute change of light intensity. It depends heavily on the stability of the instrument and is sensitive to environmental factors, such as temperature and vibration. Recently, some researchers have equated the optical cavity of IBBCEAS

to multi-pass cell, and then the gas concentration is determined according to the DOAS retrieval algorithm (Herman et al., 2009;Hoch et al., 2014;Horbanski et al., 2019;Meinen et al., 2010;Platt et al., 2009;Thalman and Volkamer, 2010). Because DOAS retrieval uses the narrow-band differential absorption characteristics of trace gas to quantify the gas concentration, the IBBCEAS system based on DOAS retrieval is insensitive to the broadband change of light intensity. The key point of this technology is to determine the effective

absorption path length, so as to correct the absorption cross section. The most common method to correct the effective absorption path length uses the measured optical density (Hoch et al., 2014). However, in this way, as with conventional IBBCEAS, the retrieved gas concentration will be affected by fluctuations in the intensity of the light source. There are also attempts to calculate the effective absorption optical path with known absorption of $O_4$ gas, but the measurement accuracy is limited (Thalman and Volkamer, 2010;Herman

et al., 2009). Recently, Horbanski et al. (2019) used an iterative method to calculate the effective absorption optical path, and developed a $NO_2$ instrument, indicating the effectiveness of the method (Horbanski et al., 2019).

This paper describes the improvement of the IBBCEAS system for the simultaneous detection of atmospheric HONO and $NO_2$. The concentration of HONO and $NO_2$ are determined by multiple iterations combined with

the DOAS retrieval algorithm. The instrument can eliminate the influence of the broadband change of light intensity by using the iterative algorithm and prevent the instability of the light source and mechanical vibration. Kalman filtering technology is an effective post-processing technology for gas concentration measurement. It has been applied to IBBCEAS (Fang et al., 2017) and laser absorption spectroscopy (Leleux et al., 2002;Wu et al., 2010). In this work we have also applied the Kalman filtering technique to improve the

detection sensitivity of the system under the same integration time. To our knowledge, the Kalman filtering technique is first used for HONO and $NO_2$ measurements. The Our instrument's capability of making fast high-sensitivity measurements of HONO and $NO_2$ is of great significance for understanding the source of HONO and studying its role in atmospheric chemistry.

## 2 System and principle

### 2.1 Theory of IBBCEAS



The basic idea of the IBBCEAS system based on an iterative algorithm is to use a high finesse optical cavity to increase the effective absorption light path, thereby improving the detection sensitivity of the instrument, and using the DOAS retrieval algorithm to determine the gas concentration (Hoch et al., 2014;Herman et al., 2009;Meinen et al., 2010;Platt et al., 2009;Thalman et al., 2015;Thalman and Volkamer, 2010). For this

system, the cavity-enhance optical density $D_{CE}(\lambda)$ is defined as follows (Horbanski et al., 2019):

$$D_{CE}(\lambda) = \bar{L}_{eff}(\lambda) \cdot \left( \sum_i c_i \cdot \sigma_i(\lambda) + \varepsilon_b(\lambda) \right) \tag{1}$$

where $c_i$ is the concentration of the gas species $i$, $\sigma_i(\lambda)$ is the absorption cross-section of the gas at wavelength $\lambda$, $\varepsilon_b(\lambda)$ is the broadband extinction caused by Rayleigh scattering and Mie scattering, and $\bar{L}_{eff}(\lambda)$ is an effective path length.

The DOAS evaluation determines the gas concentrations based on:

$$\ln\left( \frac{I_0(\lambda)}{I(\lambda)} \right) = \bar{L}_{eff} \cdot \left[ \Sigma \sigma_i'(\lambda) c_i + \text{ polynomial } \right] \tag{2}$$

Here, $I_0(\lambda)$ is the intensity of light passing through the cavity without gas absorption, $I(\lambda)$ is the intensity of light passing through the cavity with gas absorption, $\sigma_i'(\lambda)$ is the differential part of the absorption cross section of the gas, and the polynomial term in Eq. (2) represents the broadband spectral structure in the

measurement spectrum other than the differential absorption part. In traditional DOAS, the optical path length $\bar{L}_{eff}$ is a constant as determined by the physical distance, whereas in IBBCEAS it is not a constant and has a dependence on the optical density (Platt et al., 2009).

The gas concentrations of IBBCEAS measurements can be obtained by using the DOAS evaluation. Here, the wavelength-dependent effective absorption optical path $\bar{L}_{eff}(\lambda)$ is calculated by an iterative algorithm.

The implementation of the algorithm is described in Section 3.3.

## 2.2 Optical Layout

The IBBCEAS system in this study was developed based on Duan et al. work (2018) [25]. The optical layout of the IBBCEAS system is shown in Figure 1. The light source of the instrument uses a near-ultraviolet LED (LEDengin) with a center wavelength of 368 nm. In order to ensure the stability of the LED light intensity,

the LED is mounted to a Peltier device with heat sink, and the temperature of the LED is stabilized at 20°C ± 0.1°C via a thermistor temperature sensor and a PID controller. The light from the LED is coupled to a long 68 cm optical cavity through the achromatic lens (Edmund). The optical cavity is composed of two high-reflectivity mirrors and a perfluoroalkoxy alkanes (PFA) tube with an inner diameter of 22 mm. The high-reflectivity mirror (Layertec) is installed in the adjustment frame at both ends of the optical cavity. The light

transmitted through the optical cavity is filtered by a band-pass filter and then focused by an off-axis parabolic





mirror (Edmund), and finally coupled to one end of an optical fiber (600 μm, Ocean Optics). The other end of the optical fiber is connected to a spectrometer (QE65000pro, Ocean Optics), and the spectrometer is used to collect the corresponding spectral signal.

In order to prevent the reflectivity degradation during the measurement due to adsorptions of aerosol or organic species onto the mirror surfaces, the surfaces of the two high-reflectivity mirrors are purged with high-purity nitrogen (99.999%). The flow rates of the purge are controlled by two mass flow controllers at 0.1 standard liter per minute. The sampling tube of the instrument consists of a PFA tube with an outer diameter of 6 mm. A 0.2-μm PTFE filter membrane is connected to the inlet of the sampling port to prevent particles from entering the optical cavity. A diaphragm pump draws the ambient air into the instrument through the sampling tube at a flow rate of 6 standard liters per minute. The ambient air enters the system and is divided into two paths through a three-way PFA joint, one of which is discharged from the air outlet, and the other airflow enters the cavity and at a flow rate of 1.2 standard liters per minute using a mass flow controller. The use of this sampling gas path can ensure that the residence time of the atmosphere air in the sampling tube can be shortened. It does so by increasing the total sampling flow rate while maintaining a fractional constant flow rate of the cavity, thereby reducing the secondary generation and loss of HONO in the sampling tube. The software control interface is programed with LabVIEW language to ensure the orderly operation of the mass flow controller and spectrometer in the process of instrument operation.

## 3 Results and analysis

### 3.1 Determination of Mirror Reflectivity

Since the absorption optical path of the gas in the optical cavity is related to the reflectivity of mirrors, it is necessary to determine the reflectivity of the high-reflectivity mirrors before the gas concentration is retrieved. According to the method mentioned by Washenfelder et al. (2008)(Washenfelder et al., 2008), the Rayleigh scattering difference between nitrogen and helium is used to determine the wavelength-dependent reflectivity of the cavity mirrors ( $R(\lambda)$ ):

$$R(\lambda) = 1 - d_0 \frac{\left( \frac{I_{N_2}(\lambda)}{I_{He}(\lambda)} \varepsilon_{Ray}^{N_2}(\lambda) \right) - \left( \varepsilon_{Ray}^{He}(\lambda) \right)}{1 - \left( \frac{I_{N_2}(\lambda)}{I_{He}(\lambda)} \right)} \qquad (3)$$

Here, $d_0$ is the cavity length, $\varepsilon_{Ray}$ is the extinction caused by Rayleigh scattering, $I_{N_2}$ and $I_{He}$ are the light intensity when the cavity is filled with nitrogen and helium, respectively.

The accuracy in determining the mirror reflectivity will affect the accuracy of subsequent gas concentration measurements. Therefore, for achieving high measurement accuracy, it is necessary to ensure that the ambient temperature of the instrument is stable. Then the high purity nitrogen (99.999%) and high purity Helium are introduced into the optical cavity in turn, and the corresponding spectra are recorded after the spectrum



become stable after gas filling. The wavelength-dependent mirror reflectivity is calculated by substitute the ratio of the recorded nitrogen spectrum intensity and helium spectrum intensity into the Eq. (3). The resulting dependence of the mirror reflectivity on wavelength is shown in Figure 2. The red line is the spectrum
measured when the cavity is flushed with nitrogen; the black line is obtained when the cavity is filled with helium. It can be seen that the reflectivity of the mirrors is ~0.99980 at the wavelength of 368.2 nm.

### 3.2 Calibration of Effective Cavity Length

Considering the effect of mixing between the actual atmosphere gas in the cavity and the nitrogen purge gas within a narrow space in front of the cavity mirrors during measurements, the effective cavity length becomes
shorter than its physical length. The calibration experiments are needed to determine the effective cavity length ($d_{eff}$). We determine the value of $d_{eff}$ by measuring the effective concentration of a $NO_2$ gas mixture, with and without mirror purge gas to the cavity mirrors. The $NO_2$ gas sample is made by mixing a cylinder gas with a nominal 10 ppm $NO_2$ and zero air in a Teflon (FEP) gas bag, which has low adsorption property. Then the $NO_2$ gas mixture in the FEP gas bag is injected into the IBBCEAS system and measured by the
iterative algorithm. The inlet flow rate of $NO_2$ was 1 standard liter per minute, and the purging flow rate at both ends of the optical cavity was 0.1 standard liter per minute. Open and close the purge gas intermittently during the operation of the instrument and record the spectrum at the corresponding time. The $NO_2$ concentration is retrieved according to the spectra measured during this period. Figure 3 shows the change of the appearing $NO_2$ concentration during purge on and off. The effective concentration of $NO_2$ become
higher after purge off, and the concentration of $NO_2$ is returned to the previous value again after purge on. This indicates that the mirror reflectivity and $NO_2$ gas mixture before and after purge off remain unchanged, and the concentration of $NO_2$ are relatively stable during purge on and off.  The average concentration of $NO_2$ measured at purge on is 62.64±0.32 ppb, the average concentration of $NO_2$ measured after purge off is 70.92±0.19 ppb. For the cavity with physical length ($d_0$) of 68 cm, $d_{eff}$ can be calculated according to the
equation (4). The calculated $d_{eff}$ is 60.06 cm.

$$d_{eff} = d_0 \times \frac{[NO_2]_{\text{purge on}}}{[NO_2]_{\text{purge off}}} \tag{4}$$

### 3.3 Determination of Effective Absorption Optical Path and Concentration Retrieval

For traditional DOAS fitting, the retrieval result is not affected as long as there is no change of narrow-band structure, the DOAS system is insensitive to the variation of absolute light intensity. The optical absorption
path length of a usual DOAS system is constant. However, in the IBBCEAS system, the effective optical absorption path length is strongly dependent on the wavelength, which is related to the wavelength-dependency of the mirror reflectivity and intra-cavity absorption. For conventional IBBCEAS, the wavelength dependence problem is solved by converting the optical density to extinction absorption, but this method will seriously depend on the absolute stability of the light intensity. For IBBCEAS system based on


iterative algorithm, the absorption optical path is not equal to the average optical path ($\overline{L}_0(\lambda)$), so $\overline{L}_0(\lambda)$ needs to be corrected. Previous studies have tried to calculate the effective absorption optical path by using the concentration of known gases, such as $O_4$. However, the measurement accuracy is limited when the relative variation of the wavelength dependence of the path length is corrected with a single wavelength, and the peak wavelength of $O_4$ in retrieval band is at 380nm and the absorption cross-section of $O_4$ at 380nm is

three times weaker than that at 477 nm, the error may occur in retrieving the slant column density of $O_4$. According to the description of Horbanski et al. (2019), the scalar correction factor is not suitable for strong differential absorbers because their absorption band distortion can only be corrected by wavelength-resolved. Recently, Horbanski et al. (2019) proposed an iterative method to calculate the wavelength-dependent effective absorption optical path length. Through multiple iterative retrieval, the effective absorption optical

path can be finally determined. Because the filter membrane is added to the front end of the sampling port of the system in this work, the influence of gas absorption and Rayleigh scattering is only considered in the retrieval of concentration. According to Horbanski et al. (2019), the cavity-enhanced $D_{CE}(\lambda)$ of the system can be defined as:

$$D_{CE}(\lambda) = \ln\left(\frac{\int_0^\infty I_{in0}(n)dn}{\int_0^\infty I_{in}(n)dn}\right) = \ln\left(\frac{\int_0^\infty I_{in}(0) \cdot e^{-(1-R(\lambda)+\varepsilon_R(\lambda)d_0)n}dn}{\int_0^\infty I_{in}(0) \cdot e^{-(1-R(\lambda)+\varepsilon_R d_0 + \sigma(\lambda)\cdot c \cdot d_0)n}dn}\right) = \ln\left(1 + \frac{\sigma(\lambda)\cdot c \cdot d_0}{1-R(\lambda)+\varepsilon_R(\lambda)d_0}\right) \quad (5)$$

here, n indicates the n-th intra-cavity reflection. The total transmission can be considered as a sum of transmissions of all the individual intra-cavity sub beams of consecutive mirror reflections.

However, there is a nonlinear relationship between the effective absorption optical path and the optical density, which can be defined as follows:

$$\overline{L}_{eff}(\lambda) = \overline{L}_0(\lambda) \cdot \frac{D_{CE}(\lambda)}{e^{D_{CE}(\lambda)}-1} = \frac{d_{eff}}{d_0} \cdot \frac{d_0}{1-R(\lambda)+d_0 \cdot \varepsilon_R(\lambda)} \cdot \frac{D_{CE}(\lambda)}{e^{D_{CE}(\lambda)}-1} \quad (6)$$

Therefore, if we know the optical density, we can correct the effective absorption optical path, and we can continuously approach the real effective absorption optical path by multiple iterative retrieval methods. Horbanski et al. (2019) have made a detailed derivation; here is only a brief introduction, mainly divided into the following steps:

     1. Assuming that the concentration of HONO and $NO_2$ is known, the optical density can be calculated

according to equation (5).

     2. Combined with the optical density and lens reflectivity calculated in (1), the effective absorption optical path is calculated according to equation (6).

     3. Using the DOAS method to fit $D_{CE, meas}(\lambda)$ and $\overline{L}_{eff}(\lambda) \cdot \sigma_i(\lambda)$, new HONO and $NO_2$ concentration values are obtained.

4. The HONO and $NO_2$ concentrations obtained in Step 3 are replaced into Step 1 to recalculate the optical density, so the Step 1, Step 2, and Step 3 are repeated, until the change of HONO and $NO_2$ concentrations to their values of the previous iteration become less than an allowable tolerance range, a stop condition for



iteration is that the concentration difference between two retrieval is less than the fit error. The final retrieval results of HONO and NO$_2$ concentrations are obtained. The retrieval steps are shown in Figure 4.

All the data processing is based on DOASIS software (Kraus, 2006). The algorithm takes the high-resolution cross sections of HONO (Stutz et al., 2000), NO$_2$ (Voigt et al., 2002), and O$_4$ (Greenblatt et al., 1990) as the input, and then convolutes these high-resolution cross sections with the instrument functions of 0.49 nm FWHM. The fitting range of the spectrum is from 363 nm to 388 nm. Figure 5 shows the change of effective optical path length in iterative algorithm retrieval, where L1, L2, L3, L4 and L5 are the corresponding

absorption optical path from zero iteration to the fourth iteration, respectively. The effective absorption optical path length is finally converged as the number of iterations increases. Based on the effective absorption optical path length at the final iteration, the real concentration can be obtained according to the DOAS fitting. Figure 6 shows the final HONO and NO$_2$ concentrations obtained by using the iterative algorithm mentioned above in the actual large spectrum retrieval. The HONO and NO$_2$ concentrations

obtained by the final fitting are 0.78 ppbv and 29.18 ppbv, respectively.

### 3.4 Detection Limit and Uncertainty of the System

The Allan deviation is often used to calculate the sensitivity and stability of the system. Ideally, the detection sensitivity of the system can be improved by averaging successive measurements or by integrating the signal for longer time. However, actual measurement processes are affected by instrument drifts and other noise

contributions. The system can only achieve an optimal detection sensitivity before slow-varying factors become dominant. Therefore, Allan deviation can be used to describe the overall performance and stability of the system. Zero air was introduced into the optical cavity at a flow rate of 0.1 standard liters per minute, and 10,000 spectra were continuously recorded. The integration time of each spectrum is 3 s, and each spectrum is fitted using the DOAS algorithm to obtain HONO and NO$_2$ concentrations. Then the Allan

deviation is calculated according to Equation (7):

$$\delta_A^2(\tau) \ = \frac{1}{2(m-1)}\sum_{k=1}^{m-1}\left(y_{k+1}(\tau)-y_k(\tau)\right)^2 \tag{7}$$

where m is the number of time series, and $y_k(\tau)$ is the average concentration during a time interval of $\tau$. Figure 7 shows the variation of Allan deviation of the system with integration time. Under the integration time of 60 s, the detection limits (2δ) of HONO and NO$_2$ are 0.08 ppbv and 0.14 ppbv, respectively. Table 1

shows the detection limits of HONO and NO$_2$ measured by different IBBCEAS instruments reported in literatures. Our system has higher detection sensitivity than most other instruments. The Allan deviation continuous its decrease for averaging times up to several hours. This shows that the instrument is very stable in performances.

The uncertainty of the system may be determined by a Gaussian error propagation. The uncertainty of the

system is mainly composed of contributed by the uncertainties in absorption cross sections of the spectral features, mirrors' reflectivity, effective cavity length, and temperature and pressure in the cavity. The uncertainty of the mirrors' reflectivity is 5%. The uncertainty of the effective cavity length is 3%. The



uncertainty of the temperature and pressure in the cavity is 1%. The uncertainty of the fit retrieval is 4%. According to the literature, the uncertainty of the $NO_2$ and HONO absorbed cross-sections is 4%

(Voigt et al., 2002) and 5% (Stutz et al., 2000), respectively. Therefore, the total uncertainty of the instrument is about 8.1% for $NO_2$ measurement and 8.7% for HONO measurement.

**3.5 Effect of Light Intensity Fluctuation**

In order to verify the insensitivity of IBBCEAS based on iterative algorithm to the broadband change of light intensity, we carried out the experiment of light intensity fluctuation. By adjusting the angle of the off-axis

parabolic mirror to change the intensity of the spectrum, the original lamp spectrum is attenuated by 95%, 90%, 85%, 80% and 75% respectively. After adjusting the light intensity of the lamp spectrum every time, the actual atmospheric HONO concentration is measured. Finally, three methods are used to retrieve the measured atmospheric spectrum, the original lamp spectrum combined with iterative algorithm is used for concentration retrieval in method 1, the original lamp spectrum combined with conventional IBBCEAS

retrieval algorithm is used for concentration retrieval in method 2, the lamp spectrum after each change combined with the conventional IBBCEAS retrieval algorithm  is used for concentration retrieval in method 3. In order to ensure that the light intensity is not affected by the external environmental factors during the measurement process and to avoid the influence of the light intensity fluctuation caused by the external environmental factors on the measurement results, the measurement time of each cycle should be as short as

possible. Figure 8 shows the retrieval results of three methods under different relative light source intensity. The results show that the retrieval results of method 2 are influenced greatly by the fluctuation of light intensity, while the difference between the retrieval results of methods 1 and 3 is relatively small.  Figure 9 shows the fitting results of method 2 and method 3 after light intensity change. When the light intensity changes, the RMS of the fitting residual increases by 2 times. Although the HONO concentration can also be

fitted well after the light intensity changes in method 2, the attenuation signal of the lamp spectrum is considered to be caused by HONO absorption, which leads to the high measurement result. Method 3 uses the lamp spectrum after the change of the light intensity to participate in the retrieval, which ensures the absolute stability of the light intensity. But the lamp spectrum after light intensity change cannot be obtained in time in actual measurement. Method 1 always uses the original lamp spectrum to participate in the retrieval,

which shows that it is independent of the fluctuation of the light intensity of the light source.

**3.6 Ambient Measurement and Compared with Conventional IBBCEAS**

Atmospheric HONO and $NO_2$ observations were carried out in the suburbs of Hefei City, Anhui Province (31.89°N, 117.17°E), from the 29[th] of September to the 1[st] of October 2019. The IBBCEAS system based on iterative algorithm was placed in a room with approximately constant temperature controlled by air

conditioner, and the sampling port was outside of the room. A conventional IBBCEAS (Duan et al., 2018) and the IBBCEAS system based on iterative algorithm were used to measure the concentrations of HONO and $NO_2$, respectively. The conventional IBBCEAS instrument was developed and reported by our research





group previously [25]. The length of the air sampling tubes for both instruments is about 2 m. The time series of HONO and $NO_2$ concentrations measured by the two instruments is shown in Figure 10. The integration

time was 1 min. The highest HONO concentration of 3.12 ppbv appeared on the evening of September 29[th], whereas the average value of HONO during the measurement period was 0.96 ppbv. The average concentration of $NO_2$ was 15.45 ppbv, and the maximum value was 49.55 ppbv. Figure 11 shows the correlation between the measurement results of the conventional IBBCEAS and the new IBBCEAS system based on iterative algorithm. The correlation coefficients ($R^2$) of HONO and $NO_2$ results obtained by the two

instruments are 0.94 and 0.99, respectively. The differences between two IBBCEAS systems are 1% and 7% for HONO and $NO_2$ measurements, respectively, which is within the measurement uncertainty of instruments discussed in Section 3.4. Although there are many inter-comparisons about HONO measurements, the differences between different instruments have always existed. Crilley et al. (2019) reported that the IBBCEAS instrument and wet chemical method have good consistency in field observations, but the absolute

concentration difference is 12%–39% (Crilley et al., 2019). The cause of the difference was not clearly identified, so more experiments on HONO comparison are needed in the future.

### 3.7 Applications of Kalman Filtering to measurement results

Allan deviation describes the relationship between the integration time of the system and the stability of the system. According to the results of the Allan deviation, the optimum integration time of the system can be

obtained, so that the system achieves the best detection sensitivity. Kalman filtering technology can further improve the detection sensitivity of the system(Leleux et al., 2002;Fang et al., 2017;Wu et al., 2010). Compared with simple moving average, Kalman filtering can deal with "lag" effect and abnormal peak value(Leleux et al., 2002). The basic idea of Kalman filtering is to obtain the predicted state of the present time based on the state of the previous time, and the predicted state of the present time is fused with the

observed state measured by the sensor to obtain the estimation of the current state (Wu et al., 2010;Leleux et al., 2002). It can be expressed as:

$$\hat{x}_k = \hat{x}_k^- + K_k \left( z_k - \hat{x}_k^- \right) \tag{8}$$

Here, $\hat{x}_k$ is the predicted value of time sequence k, $\hat{x}_k^-$ is the predicted value of time sequence k-1, $K_k$ is the Kalman gain, and $z_k$ is the measurement value of time sequence k. In this work, the variance of previous

10 values of concentration measurements are used in the Karman filtering. Figure 12 shows the measurements of HONO and $NO_2$ concentrations under zero air conditions, together with the application of the Kalman filtering. The measurement fluctuation at the integration time of 3 s was 0.33 ppbv and 0.18 ppbv for the $NO_2$ and HONO. After applying Kalman filtering, the effect of the fluctuation is reduced and the detection sensitivity at the integration time of 3 s was 0.04 ppbv and 0.07 ppbv for the $NO_2$ and HONO. The results

show that the detection sensitivity improved by a factor of 4.5. This result is comparable to the measurement sensitivity at the integration time of 60 s. Therefore, using Kalman filtering technology can enhance time resolution and reducing measurement noise. Figure 13 shows the measurement of HONO and $NO_2$



concentrations with and without Kalman filtering under ambient conditions. In order to capture the sharp change of $NO_2$ concentration, the filter's gain parameter is set to 40. The filtering results follow the change

of the measured concentration, effectively reduce the influence of noise on the concentration results, and improve the detection sensitivity of the system.

## 4 Conclusions

Here, we have developed an IBBCEAS system based on an iterative algorithm for simultaneous measuring of atmospheric HONO and $NO_2$. The effective absorption optical path length is obtained by the iterative

algorithm. The concentrations of HONO and $NO_2$ are then determined by DOAS retrieval. The iterative algorithm is insensitive to the broadband change of light intensity and has good robustness. The reflectivity of the high-reflectivity mirror is characterized by the difference in observed Rayleigh scattering between nitrogen and helium. The reflectivity of the mirror is measured to be 0.99980 at the wavelength of 368.2 nm. The effect of the cavity-mirror-protecting nitrogen purge on the effective cavity length is calibrated using a

stable $NO_2$ concentration. The detection sensitivity of the system is analyzed by Allen deviation analysis. The detection limit of the system is 0.08 ppbv ($2\delta$ value) and 0.14 ppbv ($2\delta$ value) for $NO_2$ and HONO at an integration time of 60 s, respectively. The IBBCEAS based on iterative algorithm is in good agreement with the conventional IBBCEAS system when applied to a 'proof-of-concept" atmospheric measurement for more than two days. The total uncertainty of the system is about 8.1% for $NO_2$ measurement and 8.7% for HONO

measurement.

We also utilized a Kalman filtering technique to improve the detection sensitivity of the IBBCEAS system. This helps to realize high-sensitivity and high-resolution measurements of atmospheric HONO and $NO_2$. After applying Kalman filtering technology, the detection sensitivity at an integration time of 3 s can reach the detection sensitivity which is equivalent to an integration time of 60 s without applying Kalman filtering.

The system has a good application prospect for follow-up researches of atmospheric HONO under several different platforms, such as vehicle, balloon, and airborne platforms.





**Data availability**

The data used in this study are available from the corresponding author upon request (mqin@aiofm.ac.cn).

**Author contributions**

MQ, PX, JL, and WL contributed to the conception of the study. KT, JD, and WF built the IBBCEAS instrument. KT, JD, HZ, FM, and KY performed the experiments. KT and MQ performed the data analyses. KT wrote the manuscript. MQ and YH edited and developed the manuscript.

**Competing interests**

The authors declare that they have no conflict of interest.

**Acknowledgements**

This work was supported by the National Natural Science Foundation of China (Grants No. 41875154), the National Key R&D Program of China (Grant No. 2017YFC0209400, 2016YFC0201000 and 2017YFC0209900).

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





**Table 1.** The detection limits of HONO and $NO_2$ measured by various IBBCEAS instruments.

| Time resolution (s) | HONO detection limit (ppbv,2δ) | $NO_2$ detection limit (ppbv, 2δ) | References |
|---|---|---|---|
| 20 | 4 | 14 | Gherman et al. (2008) |
| 120 | 0.6 | 2 | Wu et al. (2014) |
| 900 | 1.5 | 1.5 | Donaldson et al. (2014) |
| 600 | 0.6 | 0.6 | Scharko et al. (2014) |
| 5 | 0.35 | 0.08 | Min et al. (2016) |
| 300 | 0.4 | Not mentioned | Naksshima and Sadanaga. (2017) |
| 30 | 0.18 | 0.34 | Duan et al. (2018) |
| 300 | 0.11 | 0.42 | Jordan and Osthoff. (2020) |
| 60 | 0.08 | 0.14 | This work |


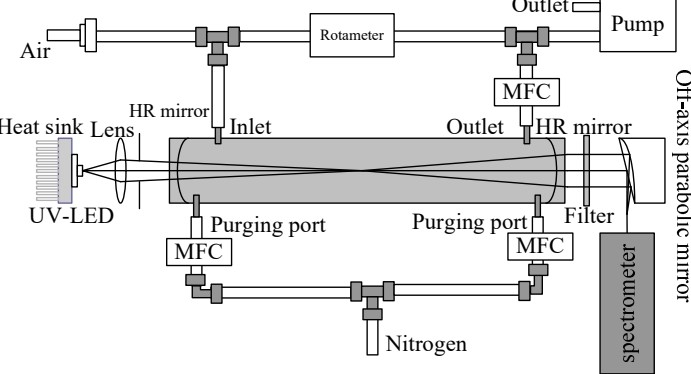


**Figure 1.** The optical layout of the IBBCEAS system. MFC: mass flow controller; HR mirror: high-reflectivity mirror.




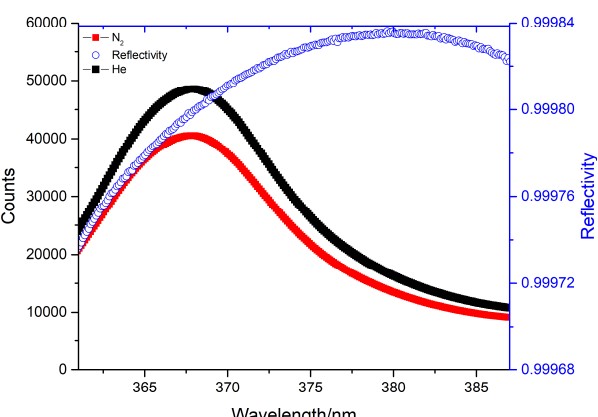

**Figure 2.** Dependences of transmission intensity when cavity is filled with $N_2$ and He gas, respectively, and the calculated cavity mirror reflectivity.


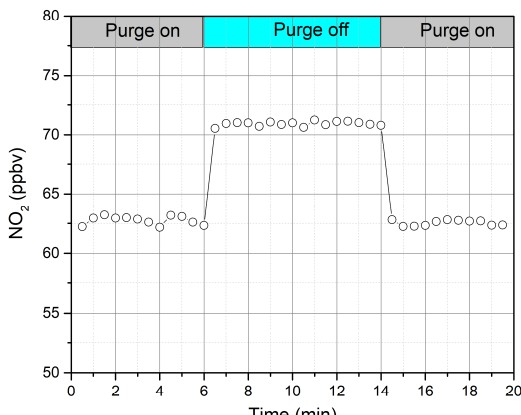

**Figure 3.** The change of effective $NO_2$ concentrations during purge on and off. The effective cavity length is calculated according to the retrieved concentrations when purging is opened and closed.







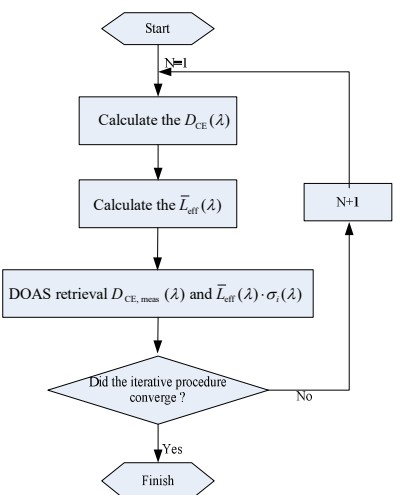

**Figure 4.** Flowchart of an iterative retrieval scheme for IBBCEAS.

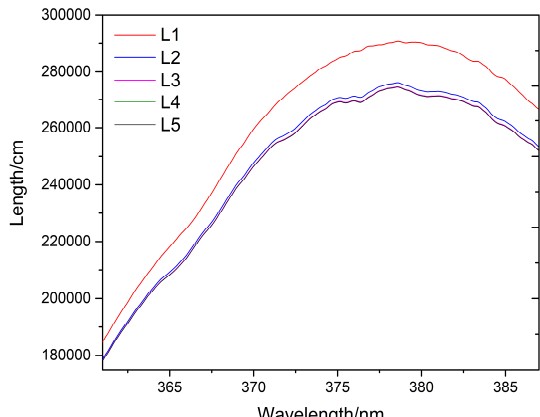

**Figure 5.** Example of the effective absorption optical path lengths calculated during consecutive iterations (L1, L2, L3,
L4 and L5) of the iterative retrieval algorithm. As the number of iterations increases, the effective absorption optical path
gradually converges.


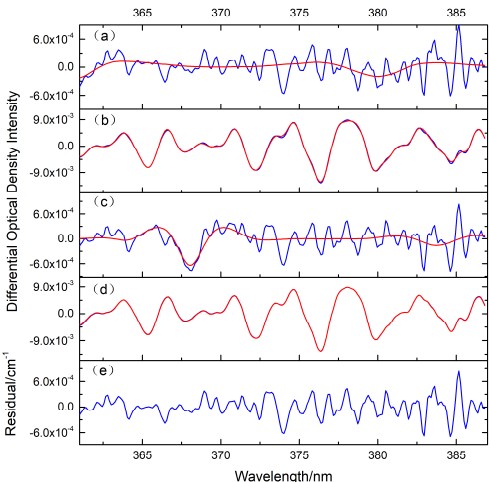

**Figure 6.** Examples of spectral retrieval of $O_4$, $NO_2$ and HONO in the measured atmosphere. (a) The red line shows the absorption spectrum of $O_4$, and the blue line shows the fitting spectrum; (b) the red line shows the absorption spectrum of $NO_2$, the blue line shows the fitting spectrum, and the retrieval $NO_2$ concentration is 29.18 ppb; (c) the red line shows the absorption spectrum of HONO, the blue line shows the fitting spectrum of HONO, and the retrieval HONO concentration is 0.78 ppb; (d) the red line shows the differential optical density line, and the blue line shows the fitting differential optical density; (e) the blue line shows the fitting residuals.

**Figure 7.** Allan deviation plots for measurements of HONO and $NO_2$. The Allan deviations of HONO and $NO_2$ are 0.08ppbv and 0.14ppbv at an integration time of 60 s, respectively.





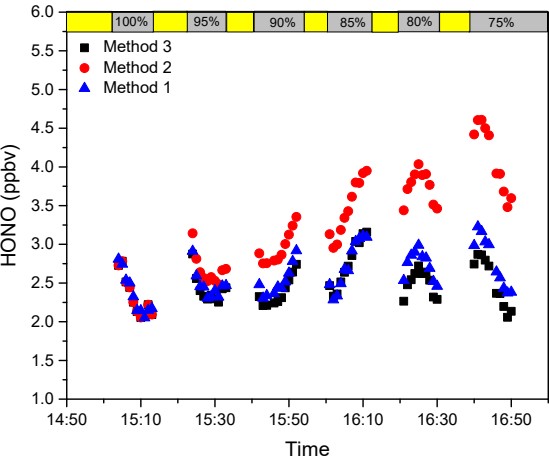


**Figure 8.** Retrieval results of three methods under different relative light source intensity. The yellow area represents the measuring time of the lamp spectrum and the gray area represents the atmospheric measurement after changing the relative light intensity of the light source.

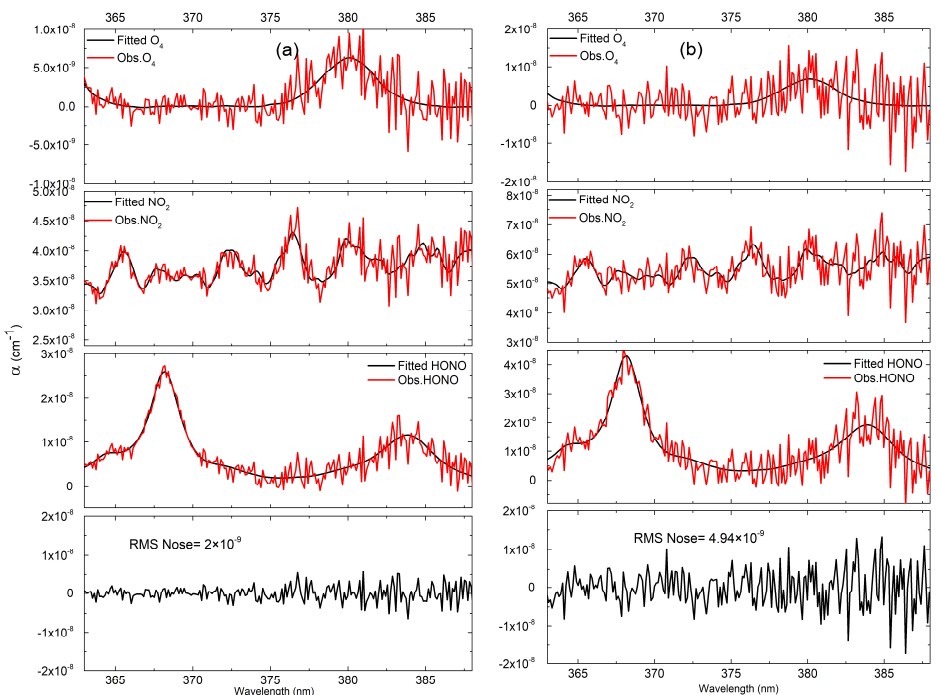


**Figure 9.** Spectral fitting results of (a) method 3 and (b) method 2 after light intensity change. The RMS of fit residuals in method 2 is $4.94 \times 10^{-9}$, The RMS of fit residuals in method 3 is $2 \times 10^{-9}$.

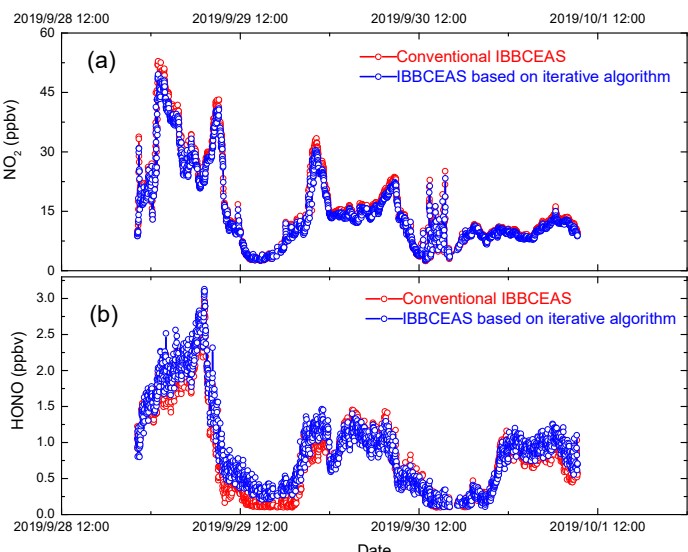

**Figure 10.** The time series of (a) $NO_2$ and (b) HONO concentrations measured by the two instruments. All data are 1 min averages. The red data point is the result of retrieval using iterative algorithm, and the blue data point is the result of retrieval using conventional IBBCEAS algorithm.

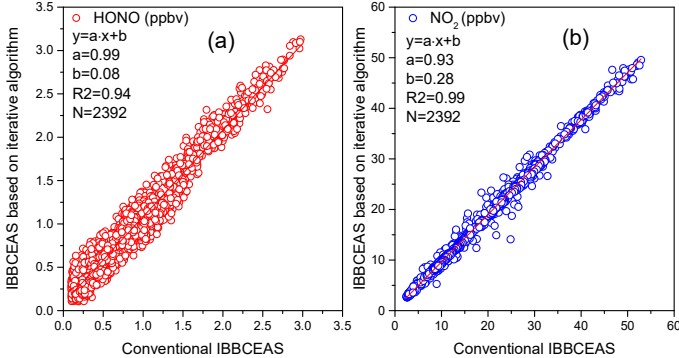

**Figure 11.** The correlation between the measurement results of the two IBBCEAS systems. (a) Correlation between the HONO concentrations determined by two IBBCEAS systems; (b) correlation between the $NO_2$ concentrations determined by two IBBCEAS systems.

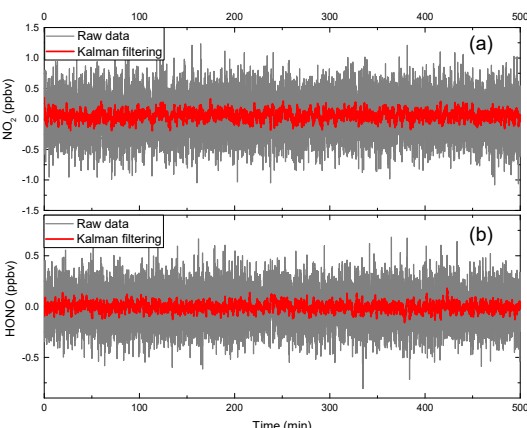


**Figure 12.** The measurement results of (a) HONO and (b) NO$_2$ with and without Kalman filtering under zero air conditions. The red line is the concentration result after Kalman filtering technology processing, gray line is the result of original concentration result.

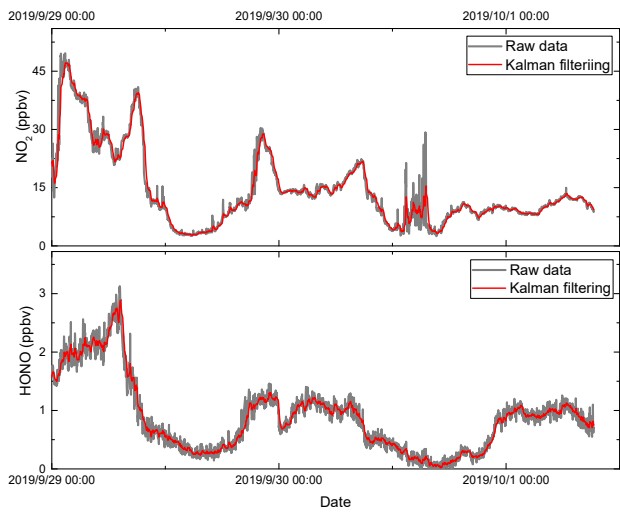

**Figure 13.** The measurement results of HONO and NO$_2$ with and without Kalman filtering under ambient conditions. The red line is the concentration result after Kalman filtering technology processing, gray line is the result of original concentration result.