# Peer review of "Simultaneous detection of atmospheric HONO and NO2 utilizing an IBBCEAS system based on an iterative algorithm"

_Atmospheric Measurement Techniques, 2020_

## Referee Comment (RC1) · Anonymous Referee #2 · 11 Aug 2020

This work proposed an improved iterative retrieval algorithm for IBBCEAS HONO and NO2 measurement and applied to real ambient situations. It can be published after sorting out following concerns and corrections.

1. As author cited a lot previous works, such as Horbanski et al. (2019); Wu et al., 2010; Leleux et al., 2002, can author be more clarity for the novelty of this work as its quite ambiguous to find out?

2. In L-96: "prevent the instability of light source" Does iterative retrieval algorithm prevent instability of light source which could included both intensity fluctuations and wavelength variations?

[Figure]

3. In L-121: "Whereas in IBBCEAS it is not a constant and has a dependence on the optical density" What quantification of the optical density can effect? Is it true for the most of ambient measurements?

4. In L-135: Please specify the bandpass filter.

5. In L-165: What is the Helium purity?

6. The Eq. 5 and 6 were both simplified, should be explained more clearly.
* * *

---

## Referee Comment (RC2) · Anonymous Referee #3 · 29 Sep 2020

Please check the attachment or the following comments:

Title: Simultaneous detection of atmospheric HONO and NO2 utilizing an IBBCEAS system based on an iterative algorithm Author(s): Ke Tang, Min Qin, Wu Fang, Jun Duan, Fanhao Meng, Kaidi Ye, Helu Zhang, Pinhua Xie, Yabai He, Wenbin Xu, Jianguo Liu, and Wenqing Liu MS No.: amt-2020-218 MS Type: Research article The authors reported an improved incoherent broadband cavity-enhanced absorption spectroscopy (IBBCEAS) system based on iterative retrieval algorithm for simultaneous detection of atmospheric nitrous acid (HONO) and nitrogen dioxide (NO2). The new algorithm is based on the concept of differential optical absorption spectroscopy (DOAS) retrieval,

which was used to obtain the effective absorption optical path length, thus IBBBCEAS system is insensitive to the fluctuation of absolute light intensity. Compared with traditional IBBCEAS, this method indicated some advantages, especially decreased the dependence on light intensity fluctuation. However, this manuscript includes some errors and uncorrected descriptions, major revision is needed to improve the paper quality. I do have some comments on the contents, interpretation, and conclusions of the work as below:

1. For detection limit, the authors used "$2\delta$", this description is very strange. The common description is "$2\sigma$", which standards for signal to noise ratio of 2.

2. Page 2, Line 69, "IBBCEAS technology has been demonstrated to applied HONO" includes grammar error, "applied" should be "apply to"

3. Page 3, lines 84 and 85, "The key point of this technology is to determine the effective absorption path length, so as to correct the absorption cross section", this description is not right, absorption cross section (of HONO or NO2) can't be corrected in this measurement and they are cited from references or database.

4. Page 3, lines 101, "The Our instrument's capability of", "The" should be deleted 5. Page 4, line 127, "The IBBCEAS system in this study was developed based on Duan et al. work (2018) [25]." developed is a redundant word and should be deleted. 6. Page 5, lines 164-165, "it is necessary to ensure that the ambient temperature of the instrument is stable." This sentence is not right, ambient temperature is variant, but you need to stabilize the gas temperature inside the optical cavity or correct the result contributed from temperature changing to decrease the uncertainty. 7. It is better to use abbreviation for "figure" (Fig.) in the manuscript, for example: Fig.1 not Figure 1 in page 4, line 128; page 6, line 169 (Fig.2 not Figure 2); page 8, line 234 (Fig.4 not Figure 4); . . . 8. "The Allan deviation continuous its decrease for averaging times up to several hours." This sentence is not right, please pay attention to the grammar and description.

9. Kalman filtering section from Pages 10 to 11. Kalman filtering can improve measurement precision but can't enhance detection limit. The authors should discriminate "detection limit" and "precision". The used detection (or measurement) sensitivity is not accurate. 10. References, all journals should use abbreviation, for example, "Atmospheric Measurement Techniques" should be "Atmos. Meas. Tech.".

11. Pages 17, lines 555-561, there is a problem in gas handling system in Fig. 1. With your current setup, the purging flow of left side will continuously dilute the sample, which will lead to extra uncertainty, please estimate this uncertainty. If you used the sampling rate of 1 SLM, the purging flow is 0.1 SLM, flow rate ratio is 10:1, can't neglect, may increase extra 9.1% uncertainty.

12. Figure 5 on Page 19, it is so hard to discriminate L1 to L5, please prepare this figure clearer for readers.

13. Figure 6 on Page 20, the authors confused the colors for the observed data and fit, the description for the red lines and blue lines in figure caption don't match the figure.

14. Figure 11 on page 22, it is better to use different colors for data and linear fit.

15. Figure 12 on page 23, it is so crowd for the figure, the text has hidden the curves in the figure.

There are other errors or wrong descriptions in this manuscript, the authors must carefully check the whole manuscript and let it more readable.

In a word, this manuscript can be considered to publish in Atmospheric Measurement Techniques (AMT) if the major revision has made to improve its quality.

Please also note the supplement to this comment:
https://amt.copernicus.org/preprints/amt-2020-218/amt-2020-218-RC2-supplement.pdf

---

## Author Comment (AC1) · 23 Oct 2020

**Response to the Reviewers' comments on the manuscript: Simultaneous detection of atmospheric HONO and NO₂ utilizing an IBBCEAS system based on an iterative algorithm**

Dear Editor and Reviewers,

Thank you very much for your letter and the comments from the referees on our paper submitted to AMT. We have checked the manuscript and revised it according to the comments.

**Comments and suggestions:**

1. As author cited a lot previous works, such as Horbanski et al. (2019); Wu et al., 2010; Leleux et al., 2002, can author be more clarity for the novelty of this work as its quite ambiguous to find out?
**Response**

Following the reviewer's suggestion, we added the sentence about the novelty of this work in Line 92-96 and Line 102-105.

**Comments and suggestions:**

2. In L-96: "prevent the instability of light source" Does iterative retrieval algorithm prevent instability of light source which could include both intensity fluctuations and wavelength variations?
**Response**

The core of the iterative algorithm is to use DOAS retrieval to detect the concentration of the gas. DOAS retrieval algorithm does not depend on the broadband variation of light intensity, the broadband spectral structure with slow change with wavelength is removed by digital filtering, and the fast change part of the remaining spectrum is fitted to obtain the concentration of gas, so the system is insensitive to the broadband change of light intensity. For wavelength variations such as spectra shift and squeeze, it has been considered in traditional DOAS retrieval (please see Stutz et al. (1996)).

**Comments and suggestions:**

3. In L-121: "Whereas in IBBCEAS it is not a constant and has a dependence on the optical density" What quantification of the optical density can affect? Is it true for the most of ambient measurements?
**Response**

According to Lambert-Beer's law, we can describe the change of light intensity after passing through an absorption path by using equation, after the corresponding transformation, we can get the defined optical density (Eq. (1)). The measured light consists of photons with different reflection times, so it will experience different distances in the optical cavity. When there is an absorber in the cavity, the effective absorption optical path will decrease nonlinearly. But if all absorbers have sufficiently wavelength-dependent absorption structures, a DOAS retrieval could be used to iteratively calculate optical density and thus determine effective absorption optical path.

**Comments and suggestions:**

4. In L-135: Please specify the bandpass filter.
**Response**

We have specified the bandpass filter in Line 135.
**Comments and suggestions:**

5. In L-165: What is the Helium purity?
**Response**

We have added the Helium purity in Line 165.
**Comments and suggestions:**

6. The Eq. 5 and 6 were both simplified, should be explained more clearly.
**Response**

Thank you for your suggestion. We reexplained the equations.

---

## Author Comment (AC2) · 23 Oct 2020

**Response to the Reviewers' comments on the manuscript: Simultaneous detection of atmospheric HONO and NO$_2$ utilizing an IBBCEAS system based on an iterative algorithm**

Dear Editor and Reviewers,

Thank you very much for your letter and the comments from the referees on our paper submitted to

AMT. We have checked the manuscript and revised it according to the comments.

**Comments and suggestions:**

1. For detection limit, the authors used "$2\delta$", this description is very strange. The common description is "$2\sigma$", which standards for signal to noise ratio of 2.

**Response**

Thank you very much. We have revised it according to the reviewer's comment.

**Comments and suggestions:**

2. Page 2, Line 69, "IBBCEAS technology has been demonstrated to applied HONO" includes grammar error, "applied" should be "apply to"

**Response**

We have revised it according to the reviewer's comment.

**Comments and suggestions:**

3. Page 3, lines 84 and 85, "The key point of this technology is to determine the effective absorption path length, so as to correct the absorption cross section", this description is not right, absorption cross section (of HONO or NO$_2$) can't be corrected in this measurement and they are cited from references or database.

**Response**

Thanks for your suggestion. We rewrote this sentence "The key point of this technology is to determine the effective absorption path length, so as to determine the gas concentration by using DOAS retrieval."

**Comments and suggestions:**

4. Page 3, lines 101, "The Our instrument's capability of", "The" should be deleted

**Response**

We have revised it according to the reviewer's comment.

**Comments and suggestions:**

5. Page 4, line 127, "The IBBCEAS system in this study was developed based on Duan et al. work (2018) [25]." developed is a redundant word and should be deleted.

**Response**

We have revised it according to the reviewer's comment.

**Comments and suggestions:**

6. Page 5, lines 164-165, "it is necessary to ensure that the ambient temperature of the instrument is stable." This sentence is not right, ambient temperature is variant, but you need to stabilize the gas temperature inside the optical cavity or correct the result contributed from temperature changing to decrease the uncertainty.

**Response**

We have revised it according to the reviewer's comment.

**Comments and suggestions:**

7. It is better to use abbreviation for "figure" (Fig.) in the manuscript, for example: Fig.1 not Figure 1 in page 4, line 128; page 6, line 169 (Fig.2 not Figure 2); page 8, line 234 (Fig.4 not Figure 4); . . .

**Response**

We have revised it according to the reviewer's comment.

**Comments and suggestions:**

8. "The Allan deviation continuous its decrease for averaging times up to several hours." This sentence is not right, please pay attention to the grammar and description.

**Response**

We have revised it according to the reviewer's comment.

**Comments and suggestions:**

9. Kalman filtering section from Pages 10 to 11. Kalman filtering can improve measurement precision but can't enhance detection limit. The authors should discriminate "detection limit" and "precision". The used detection (or measurement) sensitivity is not accurate.

**Response**

We have revised it according to the reviewer's comment.

**Comments and suggestions:**

10. References, all journals should use abbreviation, for example, "Atmospheric Measurement Techniques" should be "Atmos. Meas. Tech.".

**Response**

We have revised it according to the reviewer's comment.

**Comments and suggestions:**

11. Pages 17, lines 555-561, there is a problem in gas handling system in Fig. 1. With your current setup, the purging flow of left side will continuously dilute the sample, which will lead to extra uncertainty, please estimate this uncertainty. If you used the sampling rate of 1 SLM, the purging flow is 0.1 SLM, flow rate ratio is 10:1, can't neglect, may increase extra 9.1% uncertainty.

**Response**

We have considered the effect of adding purging flow in section 3.4.

**Comments and suggestions:**

12. Figure 5 on Page 19, it is so hard to discriminate L1 to L5, please prepare this figure clearer for readers.

**Response**

We have revised it according to the reviewer's comment.

**Comments and suggestions:**

13. Figure 6 on Page 20, the authors confused the colors for the observed data and fit, the description for the red lines and blue lines in figure caption don't match the figure.

**Response**

We have revised it according to the reviewer's comment.

**Comments and suggestions:**

14. Figure 11 on page 22, it is better to use different colors for data and linear fit.

**Response**

We have revised it according to the reviewer's comment.

**Comments and suggestions:**

15. Figure 12 on page 23, it is so crowd for the figure, the text has hidden the curves in the figure.

**Response**

We have revised it according to the reviewer's comment.

---

## Author Response (AR2)

**Response to the comments on the manuscript: Simultaneous detection of atmospheric HONO and NO₂ utilizing an IBBCEAS system based on an iterative algorithm**

Dear Editor,

Thank you very much for your comments on our paper submitted to AMT. We have checked the manuscript and revised it according to the comments.
* * *
**Comments and suggestions:**

1. Eq 4, the $NO_2$ should not be in italic format as it is a chemical formula.

**Response**

Thank you very much, we have revised it.

**Comments and suggestions:**

2. Eq 9, the notation for sigma may need to be changed consistently as commented by referee #2.

**Response**

Thank you very much, we have revised it.

**Comments and suggestions:**

3. The reference list needs proofread, some format errors still existed.

**Response**

Thank you very much, we have proofread the reference list.

[revised manuscript text omitted]